# Impact of Different Operational Definitions of Sarcopenia on Prevalence in a Population-Based Sample: The Salus in Apulia Study

**DOI:** 10.3390/ijerph182412979

**Published:** 2021-12-09

**Authors:** Luisa Lampignano, Ilaria Bortone, Fabio Castellana, Rossella Donghia, Vito Guerra, Roberta Zupo, Giovanni De Pergola, Marta Di Masi, Gianluigi Giannelli, Madia Lozupone, Francesco Panza, Heiner Boeing, Rodolfo Sardone

**Affiliations:** 1National Institute of Gastroenterology “Saverio de Bellis”, Research Hospital, Castellana Grotte, 70013 Bari, Italy; ilariabortone@gmail.com (I.B.); fabio.castellana@irccsdebellis.it (F.C.); rossydonghia@gmail.com (R.D.); vito.guerra@irccsdebellis.it (V.G.); zuporoberta@gmail.com (R.Z.); gdepergola@libero.it (G.D.P.); marta.dimasi@irccsdebellis.it (M.D.M.); gianluigi.giannelli@irccsdebellis.it (G.G.); f_panza@hotmail.com (F.P.); boeing@dife.de (H.B.); 2Department of Biomedical Science and Human Oncology, School of Medicine, University of Bari Aldo Moro, 70100 Bari, Italy; 3Center for Neurodegenerative Diseases and the Aging Brain, Department of Basic Medicine, Neuroscience, and Sense Organs, University of Bari Aldo Moro, 70100 Bari, Italy; madia.lozupone@gmail.com; 4Department of Molecular Epidemiology, German Institute of Human Nutrition Potsdam-Rehbruecke, 14558 Nuthetal, Germany

**Keywords:** sarcopenia, EWGSOP1, EWGSOP2, prevalence, study population, older adults

## Abstract

Background: In 2010, the European Working Group on Sarcopenia in Older People (EWGSOP1) issued its first operational definition to diagnose sarcopenia. This was updated in 2019 with a revised sequence of muscle mass and muscle strength (EWGSOP2). The aim of the study was to investigate the impact of these different operational definitions on sarcopenia prevalence in a representative population-based sample. Methods: For each algorithm, the prevalence of sarcopenia-related categories was calculated and related to sociodemographic and lifestyle variables, anthropometric parameters, and laboratory biomarkers. The present analysis used data from the Salus in Apulia Study (Italy, 740 subjects, mean age 75.5 ± 5.9 years, 54% women). Results: The application of the EWGSOP1 adapted algorithm resulted in 85% [95% confidence intervals (CI): 82–88%] non-sarcopenic subjects, 10% (95% CI: 8–12%) pre-sarcopenic subjects, and 5% (95% CI: 3–7%) sarcopenic/severe sarcopenic subjects. The sarcopenia-related categories were inversely related to weight and body mass index (BMI), particularly in overweight/obese subjects, and these categories showed favorable metabolic biomarkers. The EWGSOP2 algorithm yielded 73% (95% CI: 69–76%) non-sarcopenic subjects, 24% (95% CI: 21–27%) probably sarcopenic subjects, and 4% (95% CI: 2–5%) sarcopenic subjects. Conclusions: The present study identified BMI as a potential confounder of the prevalence estimates of sarcopenia-related categories in population-based settings with different EWGSOP operational definitions.

## 1. Introduction

Sarcopenia is a progressive, generalized skeletal muscle disorder involving the combination of loss of muscle mass and loss of muscle function and/or muscle strength, as well as loss of muscle performance [1]. In September 2016, sarcopenia was recognized as a disease entity and assigned an ICD-10-CM (M62.84) code [1]. This disorder particularly affects older adults, and the public health burden is substantial, since people over the age of 65 already account for 13% of the global population. The number of older people will substantially increase with time, reaching an estimated 2 billion people by 2050 [2]. It has consistently been shown that various adverse health-related outcomes are associated with sarcopenia, including falls, functional decline, frailty [3], cardiovascular diseases [4], and mortality [5].

In the near future, in order to gain a proper epidemiological description of the new disease (ICD-10 M62.84) and find adequate treatments, it will be important to reach an internationally recognized consensus as to how sarcopenia should be diagnosed. Currently, various different operational definitions and diagnostic tools have created a situation that does not allow for a clear definition of the true epidemiological size of the problem. In 2010, an important step forward in reaching a consensus for diagnosing sarcopenia was taken by the European Working Group on Sarcopenia in Older People (EWGSOP1) [6]. Recently, in 2019, details regarding case-finding, diagnosis, cutoffs, and severity determination were updated (EWGSOP2) [7]. However, this revision was associated with an important change in the sequence of diagnostic criteria and operational definitions. It assigns the discriminating role to muscle function and/or muscle strength when distinguishing non-sarcopenic, probably sarcopenic, and sarcopenic subjects, instead of muscle mass. The 2019 working group argued that the muscle strength measurement is an easy procedure and should be conducted first when heading for a diagnosis. Thus, the final role of muscle strength and muscle performance is still unclear and needs to be further explored. Therefore, all of these changes proposed by EWGSOP2 may lead to a lack of agreement with EWGSOP1, hypothesizing important changes in the prevalence of sarcopenia in different populations.

A further aspect of sarcopenia regards the role of an increased body mass index (BMI) over the course of one’s life, and the unfavorable change in body composition. Usually, in advanced age, muscle mass starts to decrease and fat mass increases, showing a shift from subcutaneous to visceral fat. This has been labeled a new entity called sarcopenic obesity [8]. Sarcopenic obesity is gaining a lot of interest in the medical community, and in clinical practice the number of studies focusing on sarcopenic obesity with regard to risk evaluation and treatment effects seems to outnumber studies on sarcopenia in general [9]. Furthermore, sarcopenia might result in physical frailty and low physical activity [10]. In fact, weakness (low muscle strength) and slowness (low gait speed) are components of the classical unidimensional physical frailty model [11]. In the present study, data from the general population recruited in Southern Italy for the ongoing long-term Salus in Apulia study were used were used to estimate the prevalence of sarcopenia, assuming a substantial difference in percentage terms according to the two consensuses. In addition, we explored how sarcopenia-related categories may be related to sociodemographic and lifestyle variables, anthropometric parameters, and laboratory biomarkers.

## 2. Materials and Methods

### 2.1. Study Population and Design

The present study included individuals from the Salus in Apulia Study (MICOL Study plus GreatAGE Study), an ongoing population-based prospective cohort comprising 2192 individuals aged 65 years or older and resident in Castellana Grotte, a town located near Bari, Puglia, in the southeast of Italy. The Salus in Apulia Study focused on the lifestyle sequence in older age, considering diet, frailty, and other age-related impairments and diseases [12]. The Institutional Review Board of the National Institute of Gastroenterology “S. De Bellis” issued prior approval of the Salus in Apulia Study, measurements, and data collections. The present study adhered to the Standards for Reporting Diagnostic Accuracy Studies (STARD) guidelines [Available online: http://www.stard-statement.org/ (accessed on 30 September 2021)] and the Strengthening the Reporting of Observational Studies in Epidemiology (STROBE) guidelines [Available online: https://www.strobe-statement.org/ (accessed on 30 September 2021)], and was conducted in accordance with the Helsinki Declaration of 1975. All participants provided written informed consent to join the study. The Salus in Apulia Study participants underwent nutritional, lifestyle, and geriatric evaluations via questionnaires, as well as comprehensive anthropometric, functional, and laboratory assessments [12]. The present study used data from a subpopulation of the Salus in Apulia Study, including 740 older subjects (+65 years) who underwent bioelectrical impedance analysis (BIA). BIA was offered to all the Salus in Apulia Study subjects, but not all showed interest in this examination.

### 2.2. Assessment of Lifestyle-Related and Socioeconomic Characteristics

Smoking habits and education data were collected through self-report: education was assessed as the overall years of school attendance, whereas smoking was assessed as a dichotomous variable (yes/no). Physical activity was assessed through an interviewer-administered questionnaire [13,14]. In particular, subjects were asked to indicate their average level of physical activity during the past year. One of the following six response categories, which incorporate duration, frequency, and intensity of physical activity, could be chosen for each age period: (0) bedridden, (1) minimal physical activity, (2) light physical activity performed 2 to 4 h per week not accompanied by sweating (e.g., walking), (3) moderate physical activity performed 1 to 2 h per week accompanied by sweating or light physical activity not accompanied by sweating for >4 h per week, (4) moderate physical activity performed ≥3 h per week accompanied by sweating, and (5) physical exercise that required maximal strength and endurance performed regularly, several times per week [13,14].

### 2.3. Assessment of Anthropometric Parameters and Parameters of Physical Fitness

Height was measured to the nearest 0.5 cm using a wall-mounted stadiometer (Seca 711; Seca, Hamburg, Germany). Body weight was determined to the nearest 0.1 kg using a calibrated balance beam scale (Seca 711; Seca, Hamburg, Germany). BMI was calculated as weight in kilograms divided by height in meters squared (kg/m^2^). Lean mass was measured using a BIA (BIA Quantum Akern), taken as a proxy for muscle mass. From the muscle mass, several indices were derived. Skeletal muscle index (SMI) was assessed using the equation by Janssen and colleagues [15] divided for height2 (m^2^), using cutoff points of <8.87 kg/m^2^ for men and <6.42 kg/m^2^ for women, as suggested for Italy and using this device [16,17]. Appendicular skeletal muscle mass index (ASMI) was derived from the BIA measurement applying the equation by Sergi and colleagues [18] divided for height2 (m^2^), using cutoff points of <7 kg/m^2^ for men and 5.5 kg/m^2^ for women, as suggested by the EWGSOP2 panel. The five-times-sit-to-stand (FTSS) Test represents muscle strength in our study, since hand grip strength was not assessed. The FTSS Test measures the amount of time needed for a patient to stand up five times from a seated position without using his or her arms. Longer than 15 s was considered as the cutoff for low muscle strength by the EWGSOP panel [7,19]. We divided the time and circumstances needed to stand up into five categories: 0 for subjects who did not perform the test due to physical inability, 1 for those who took more than 15 s but used their arms as support, 2 for more than 15 s but without arm support, 3 for subjects who took 15 s or less with arm support, and 4 for subjects who took 15 s or less with no support. Low physical performance was assessed using the Short Physical Performance Battery (SPPB), an objective tool for measuring the physical performance status of the lower extremities [20]. The SPPB is based on three timed tasks: standing balance, walking speed, and chair sit-to-stand tests [20]. The timed results of each subtest were rescaled according to the predefined cutoff points, obtaining a score ranging from 0 (worst performance) to 12 (best performance) [20]. A cutoff value of 8 in the SPBB score was considered to indicate low physical performance, in accordance with both EWGSOP panels [6,7].

### 2.4. Algorithms for the Operational Definitions of Sarcopenia

We followed both the EWGSOP1 and EWGSOP2 operational definitions, cutoff points, and measures as much as possible using our available data. The pre-sarcopenia category was defined as a low muscle mass index without low muscle function and/or muscle strength or physical performance [6]. The sarcopenia category was defined as low SMI in addition to either low muscle strength or low physical performance. If both latter conditions were present, a subject was assigned to the severe sarcopenia category [6]. EWGSOP2 suggested a change in sequence between muscle mass and strength [7]. The category of probable sarcopenia was defined as having low muscle strength, and the category of sarcopenia was defined by the presence of both low muscle mass and low muscle strength. As with the EWGSOP1 definition, the category of severe sarcopenia was assigned when low muscle strength, low muscle mass, and low physical performance was present [7]. Cutoff points for BIA-derived low muscle mass were not directly suggested by the EWGSOP2 operational definition, which suggested the use of ASMI instead of SMI as parameter of low muscle mass. We addressed the latter suggestion by a first analysis with the cutoff points of SMI, as used in the EWGSOP1 algorithm, and a second analysis with ASMI using the cutoff points suggested by EWGSOP2.

### 2.5. Laboratory Assessments

Plasma glucose was determined using the glucose oxidase method (Sclavus, Siena, Italy), while the concentrations of plasma lipids [triglycerides, total cholesterol, high density lipoprotein (HDL) cholesterol] were quantified by automated colorimetric method (Hitachi; Boehringer Mannheim, Mannheim, Germany). Low-density lipoprotein (LDL) cholesterol was calculated by applying the Friedewald equation. Hemoglobin, insulin, creatinine, alanine amino transferase (ALT) aspartate amino transferase (AST), and gamma glutamyl transferase (GGT) were measured using automatic enzyme procedures. The Homeostatic Model Assessment of Insulin Resistance (HOMA-IR) was calculated according to the following formula: fasting insulin (microU/L) x fasting glucose (mg/dL)/405 [21].

### 2.6. Statistical Analyses

Participants’ characteristics were reported as mean ± standard deviation (M ± SD) for continuous variables, and as frequencies and percentages (%) for categorical variables. For single sociodemographic and clinical characteristics, as well as laboratory biomarkers, the trend among different sarcopenia-related categories was analyzed by linear regression models or by applying Poisson regression models where necessary. Significant *p*-values were set at <0.05, two-tailed. In a first analysis of the EWGSOP2 operational definition, we used the EWGSOP1 variables and cutoff points. In a further analysis, the variable SMI was replaced by ASMI with its cutoff points. Sarcopenia and severe sarcopenia subjects were combined into one category, due to the small numbers. The analysis across trends between the sarcopenia categories was performed for the whole study population, but also for the normal weight and the overweight/obese subjects. The latter two analyses are not shown in the tables, but significant findings are mentioned in the text. All statistical computations were performed with StataCorp. 2019 (Stata Statistical Software: Release 16. College Station, TX: StataCorp LLC).

## 3. Results

The sociodemographic and lifestyle-related characteristics of the study sample (*n* = 740) and the total cohort (*n* = 2192) are shown in Table 1. On average, individuals were 75.5 ± 5.9 years old, with a slight predominance of women (54% vs. 46%).

Table 2 shows the correlation among the variables used for the two operational definitions of sarcopenia, and the anthropometric parameters subdivided by BMI status (normal weight and overweight). The FTSS Test and SPPB were strongly correlated, but not correlated with SMI or the other anthropometric parameters in both BMI subgroups. SMI, as a variable based on anthropometry, was correlated with height and weight in both BMI subgroups, probably due to the male/female differences. The strongest correlation was seen between height and weight in the normal weight group, exemplifying the relationships among some of the anthropometric variables which also relate to fat accumulation, as seen by the lower correlation in the overweight group. In contrast to the normal weight group, a small correlation between SMI and BMI appeared in the overweight group. ASMI was very highly correlated with SMI, but also showed a correlation with BMI in the normal weight group and a moderate-to-high correlation with BMI and weight in the overweight group, in contrast to SMI.

The EWGSOP1 definition used SMI as a variable to distinguish between normal subjects and sarcopenic and pre-sarcopenic subjects, and muscle strength or physical performance to distinguish between pre-sarcopenic and sarcopenic subjects. Table 3 describes the characteristics of these three categories (normal, pre-sarcopenic, and sarcopenic/severely sarcopenic subjects), and the calculated trend across those categories ordered by the sarcopenia severity. The characteristics included sociodemographic, lifestyle-related, anthropometric, and sarcopenia-related variables, as well as the laboratory biomarkers’ measurements. The proportion of sarcopenic/severely sarcopenic subjects (13%, 95% CI: 8–20%) was much higher among the normal weight subjects compared to the total group, and much lower among the overweight subjects, only a small proportion of whom showed sarcopenia/severe sarcopenia (3%, 95% CI: 2–4%). Females were underrepresented in the pre-sarcopenic group, whereas in all groups, age showed a significant trend across the sarcopenia-related categories, with the non-sarcopenic subjects being the youngest. There was no trend across sarcopenia-related categories regarding education, smoking and physical activity. In regard to anthropometric variables, the non-sarcopenic subjects had a higher BMI with an inverse trend across the sarcopenia categories. Weight was also inversely related to the sarcopenia categories but confined to the overweight/obese group. Height was not related to sarcopenia despite being similarly correlated with SMI as weight. There was no trend across sarcopenia-related categories for muscle strength and physical performance, with pre-sarcopenic subjects showing the highest values and sarcopenic/severely sarcopenic subjects the lowest. The biomarkers glucose, HOMA-IR, and HDL cholesterol followed the trend across sarcopenia-related categories. HDL was confined to the overweight/obese group. The highest values of glucose and HOMA-IR and the lowest values of HDL cholesterol were observed in the non-sarcopenic subjects. These trends were not seen in normal weight subjects.

The EWGSOP2 definition used muscle strength as the variable to distinguish normal subjects from sarcopenic and probably sarcopenic subjects. Muscle mass was used to differentiate between the probably sarcopenic and the sarcopenic/severely sarcopenic subjects. Physical performance was used to differentiate between sarcopenic and severely sarcopenic subjects (both categories were collapsed into one in the present study). In the first analysis we used the same variables and cutoff points for muscle mass as in the EWGSOP1 algorithm. The change in the sequence between muscle mass and muscle strength, without changing the definition of muscle mass, resulted in a different distribution of the study subjects according to the sarcopenia-related categories (Table 4).

With the change of sequence, other trends across categories became significant, when compared to the previous analysis (Table 3), and sarcopenic/severely sarcopenic subjects showed the worst averages for variables used to define sarcopenia-related categories such as SMI, and the measures of muscle strength and physical performance (Table 4). Physical activity was still inversely correlated to the sarcopenia categories as with the EWGSOP1 definition. It is noteworthy that the change in sequence was not related to anthropometric variables and the previous laboratory biomarkers’ measurements (Table 4). However, total cholesterol was now positively linked to the sarcopenia categories. The EWGSOP2 proposed the use of ASMI instead of SMI as the parameter to define muscle mass. Thus, we repeated the previous analysis using ASMI and its cutoff points (Table 5).

In terms of the prevalence of sarcopenia-related categories, no substantial differences were found compared to the previous analysis using SMI (Table 4). In the new analysis, physical activity and the measures of muscle strength and physical performance were inversely related to the sarcopenia severity categories as well as to the BMI, also in the normal weight and overweight/obese subgroups. Regarding biomarkers, no significant trend was found.

## 4. Discussion

The present study compared the first and the revised EWGSOP algorithms in a community-dwelling older population. In general, in both algorithms between 4% and 5% of the population could be labeled as sarcopenic or severely sarcopenic. The most striking difference between the first and the revised algorithm regarded the prevalence of pre-sarcopenic or probably sarcopenic subjects, which was found to be higher using the EWGSOP2 algorithm. With the EWGSOP1 algorithm, we observed a confounding effect between the sarcopenia categories and weight and BMI. Sarcopenia was linked to a lower weight and BMI, but more favorable metabolic biomarkers. The EWGSOP2 algorithm using SMI resulted in non-confounding with BMI and an inverse relation with physical activity. The EWGSOP2 algorithm, using ASMI, resulted again in a confounding effect of BMI.

The change from muscle mass to muscle strength in the EWGSOP2 algorithm to diagnose sarcopenia was mostly motivated by feasibility rather than by strictly scientific arguments [7]. The main argument was based on the findings from many studies that muscle mass and muscle strength were highly correlated [22,23]. However, we did not observe this relationship. Nevertheless, since sarcopenia is expressed as functional muscle failure [1], it has also been stated that muscle strength is the most reliable measure of muscle function, rather than muscle mass [24]. Instead, in the present study only muscle strength and physical performance were highly correlated. The latter variable is used to define the severity of sarcopenia in EWGSOP2 [7]. Due to the small numbers of affected subjects (normal muscle strength but low physical performance), we could not study the effect of this variable in detail in our investigation. Nevertheless, the present study clearly showed that a weakness in muscle function and/or muscle strength was not necessarily connected with muscle mass as measured by SMI based on height2, and that the SMI, even using national cutoff points, might not always be a good marker of loss of muscle function and/or muscle strength.

The SMI seemed to be the most interesting variable in terms of a potential impact on the diagnosis of sarcopenia. In many studies, sarcopenia was defined only by low muscle mass, ignoring the full definition of the disease [8]. We found, as in other previous studies [25,26], that SMI based on height2 is correlated with height and weight, and in overweight subjects, even BMI [27]. Unlike BMI, which was extensively investigated following the ideas of Adolphe Quetelet 150 years ago and used to assess body fat, the analogous SMI does not have such a foundation, but is used similarly to BMI, with height2 the denominator for adjustment. The analogy between BMI and SMI is only justified when assuming that the relation of weight to skeletal muscle mass follows a 45-degree line over the full weight range, which is not the case. Instead, the relation is curvilinear [28]. Moreover, it has been noticed that SMI based on height2 reflects lower muscle mass in people with a lower weight [29]. In addition, another SMI has been proposed, which considers other variables for adjustment such as weight or BMI [25]. In the present study, we used the established height2 approach due to the national data available. As a consequence, higher-weight subjects also had a higher skeletal muscle mass, and thus a greater probability of exceeding the cutoff point for SMI. Thus, we observed fewer sarcopenic subjects among the overweight subjects, and an inverse trend between weight and BMI and sarcopenia status. For completeness of information, it should be stated that BMI-adjusted SMI is also recommended by the Foundation for the National Institutes of Health (FNIH) and other studies [30,31] because it correlates better with muscle functionality [32].

It is therefore questionable whether the EWGSOP1 algorithm, which applied SMI as the initial variable identifying sarcopenic subjects or subjects at risk of sarcopenia, serves its purpose when used in population-based samples without clinical screening. A further aspect of this approach was the confounding of SMI with BMI-related metabolic disturbances, that led in the present study to inverse relations with laboratory biomarkers, reflecting a distorted glucose metabolism, for example. These observations imply that sarcopenia—as a disease of muscle function and/or muscle strength—and metabolic disturbances are inversely interlinked, whereas this did not occur when changing the sequence of muscle mass and muscle strength according to the EWGSOP2 algorithm.

The revised sequence, applying muscle strength instead of muscle mass, resulted in a negative trend across the categories of sarcopenic status with physical activity and no trend with weight and BMI or metabolic biomarkers, except for a negative trend with the cardiometabolic biomarker HDL cholesterol. Thus, the EWGSOP2 algorithm using SMI satisfied a number of issues related to physical activity, but independent from BMI and BMI-related metabolic disturbances. The inverse relationship between sarcopenia status and HDL cholesterol was also seen in a recent meta-analysis [33]. The EWGSOP2 algorithm using ASMI was again confounded by BMI in the same direction as the EWGSOP1 algorithm.

In other studies, when comparing the EGWSOP1 and EWGSOP2 algorithms [34,35,36,37,38,39,40,41], often conducted in specific clinical diagnostic groups [35,38,39], different estimates were made, with a tendency to a lower prevalence of sarcopenia with the revised algorithm, inconsistent agreement on sarcopenia status, and different covariations of confounders across the algorithms. In fact, a Chinese study found that the revised algorithm was linked to BMI, in contrast to the first one, and was associated with a much lower prevalence of sarcopenia [36]. In the Italian InCHIANTI population-based study of older adults, the algorithms were also compared and associated with mortality, showing a similar sarcopenia prevalence to the findings in the present study. In that study, muscle strength was a slightly better predictor for mortality than muscle mass [40]. The InCHIANTI study also reported a low agreement between the two algorithms in terms of individual assignments [40].

The present study identified the sarcopenia-related “probably sarcopenic” or “pre-sarcopenic” categories as those in which the prevalence differed substantially between the algorithms, and in turn affected the prevalence of non-sarcopenic subjects. From a prevention standpoint, the category at risk of sarcopenia (probably sarcopenic with low muscle strength but normal muscle mass) seems to be important, but requires further consideration and clear therapeutic approaches. Currently, the prevalence of sarcopenia, including pre-conditions in population-based samples, varies depending on covariates, the components of the sarcopenia definition, and the tools employed for measurement [42,43]. A recent systematic review and meta-analysis of general population-based studies highlighted that the use of different measurement tools, cutoff points and definitions may lead to a different prevalence of sarcopenia, and the results may be difficult to interpret [44]. In addition, non-linear relations with age or other characteristics, as reported for muscle function and/or muscle strength measurements [45], make the interpretation of results regarding the prevalence of sarcopenia across studies extremely difficult, even if the same cutoff points were used, because of the differences in age distribution across the studies. As an example, it has been recently stated that, in a Norwegian community-dwelling population, using chair stands instead of grip strength more than doubled probable sarcopenia prevalence across all ages (40–84 years) [46]. This finding suggested that the two strength measures defined individuals with contradictory anthropometrics, body composition and physical function to have probable sarcopenia.

Our findings, together with those from the InCHIANTI study, showed that sarcopenia or severe sarcopenia was not a common phenomenon in non-hospitalized normal older populations (>65 years) in Italy. However, the proportion of probably sarcopenic subjects was substantial, and in nearly the same range in the two studies (24% in the present report and about 16% in the InCHIANTI study) [40]. Other studies reported an even higher prevalence of sarcopenia, and a recent meta-analysis using older definitions concluded that in non-Asian countries the prevalence of sarcopenia with BIA measurements of muscle mass was 13% (7–19%) in men and 13% (9–19%) in women [44].

The present study has several strengths, having been performed in a large prospective population-based setting. The limitations of the present study have to do with the cross-sectional design of our study. Therefore, firstly, the estimate of prevalence cannot be considered as a risk measure of sarcopenia, and secondly, the cross-sectional nature reduces the possibility of inferring causal relationships between sarcopenia and its clinical correlates. In addition, the analysis of the body composition was carried out using the BIA method, notoriously considered less reliable than the gold standard (dual-energy X-ray absorptiometry or magnetic resonance imaging). Another weakness of the present study could be the selection bias generated by the fact that our tests were conducted at a walk-in clinic, so subjects with impaired mobility could not take part. Another limitation consists of the use of 5STS to assess muscle strength in both consensus definitions, even though EWGSOP1 considered handgrip strength or knee flexion/extension or peak expiratory flow as measurements of muscle strength [6]. Moreover, we did not use specific cutoffs for the Italian older population because, to the best of our knowledge, there are no studies on this subject, unlike other populations [47].

## 5. Conclusions

In conclusion, in this population-based study, we observed a relatively low prevalence of sarcopenia and we identified BMI as a potential confounder of the prevalence estimates of sarcopenia-related categories with different EWGSOP operational definitions, which might necessitate a more elaborate modeling of the sarcopenia disease risk relation. In addition, use of the two different algorithms did not result in a substantially different percentage of subjects with sarcopenia in our study population.

## Figures and Tables

**Table 1 ijerph-18-12979-t001:** Sociodemographic and lifestyle-related characteristics of study participants. The *Salus in Apulia* Study (*n* = 740).

Variables *	Total Cohort (*n* = 2192)	Study Sample (*n* = 740)
Age (years)	74.9 ± 6.0	75.5 ± 5.9
Smoking (Yes) (%)	45 (4.5)	29 (3.9)
Education (yrs)	6.9 ± 4.3	6.9 ± 4.3
Physical activity	2 ± 1	2 ± 1
Height (cm)	157.7 ± 8.8	157.3 ± 8.6
Weight (kg)	73.1 ± 13.9	72.8 ± 13.9
BMI (kg/m^2^)	29.3 ± 4.9	29.4 ± 4.9
SMI (kg/m^2^)	8.7 ± 1.6	8.7 ± 1.6
SMI Men (kg/m^2^)	10.0 ± 1.1	9.9 ± 1.1
SMI Women (kg/m^2^)	7.6 ± 1.1	7.6 ± 1.1
ASMI (kg/m^2^)	7.2 ± 1.1	7.2 ± 1.0
ASMI Men (kg/m^2^)	7.7 ± 0.9	7.7 ± 0.8
ASMI Women (kg/m^2^)	6.7 ± 0.9	6.7 ± 0.9
FTSS Test	3.3 ± 1.2	3.4 ± 1.2
SPPB	10.3 ± 2.4	10.3 ± 2.4

* All values: Mean ± standard deviation (mean ± SD) for continuous variables and frequencies and percentage (%) for categorical variables BMI: body mass index; SMI: skeletal muscle index; ASMI: appendicular skeletal muscle index; FTSS: five-times-sit-to-stand; SPPB: Short Physical Performance Battery.

**Table 2 ijerph-18-12979-t002:** Spearman rank correlations between variables used for the two operational definitions of sarcopenia and other anthropometric parameters. The *Salus in Apulia* Study (*n* = 740).

Normal Weight Overweight	SMI	ASMI	FTSS Test	SPPB	BMI	Height	Weight
SMI	-	0.92	0.06	0.09	0.04	0.52	0.52
ASMI	0.89	-	−0.01	0.02	0.55	0.37	0.73
FTSS Test	0.01	0.05	-	0.71	0.00	0.06	0.09
SPPB	0.06	0.09	0.64	-	−0.05	0.08	0.07
BMI	0.22	0.21	−0.07	−0.05	-	−0.04	0.42
Height	0.53	0.49	0.00	0.06	−0.17	-	0.86
Weight	0.56	0.59	−0.05	−0.01	0.67	0.58	-

SMI: skeletal muscle index; ASMI: appendicular skeletal muscle index; FTSS: five-times-sit-to-stand; SPPB: Short Physical Performance Battery; BMI: body mass index.

**Table 3 ijerph-18-12979-t003:** Sociodemographic, lifestyle-related, anthropometric, sarcopenia-related, and laboratory biomarker characteristics of the study participants according to the different sarcopenia-related categories, European Working Group on Sarcopenia in Older People 2010, EWGSOP1. The *Salus in Apulia* Study (*n* = 740).

EWGSOP1	No Sarcopenia	Pre-Sarcopenia	Sarcopenia + Severe Sarcopenia	*p* *
**Frequency**	*n* = 631 85% (95% CI: 82–88%)	*n* = 72 10% (95% CI: 8–12%)	*n* = 37 5% (95% CI: 3–7%)	
**Sociodemographic and lifestyle-related variables**				
Age (years)	75.1 ± 5.7	77.7 ± 6.6	77.5 ± 6.7	<0.001
Female (%)	346 (55)	31 (43)	21 (57)	0.75
Education (years)	6.8 ± 4.2	7.1 ± 4.8	7.1 ± 5.1	0.60
Smoking (yes, %)	22 (4)	5 (7)	2 (5)	0.24
Physical activity	2 ± 1	2 ± 1	2 ± 1	0.86
**Anthropometric variables**				
Height (cm)	157.1 ± 8.6	158.5 ± 8.7	158.1 ± 9.3	0.23
Weight (kg)	74.5 ± 13.7	63.9 ± 10.7	62.2 ± 10.3	<0.001
BMI (kg/m^2^)	30.2 ± 4.9	25.4 ± 3.3	24.9 ± 3.1	<0.001
**Sarcopenia assessment**				
SMI (kg/m^2^)	8.9 ± 1.5	7.4 ± 1.2	7.1 ± 1.3	<0.001
FTSS Test	3 ± 1	4 ± 0	2 ± 1	0.26
SPPB	10 ± 2	12 ± 1	9 ± 2	0.51
**Laboratory biomarkers**				
Glucose (mg/dL)	104.9 ± 26.4	95.6 ± 17.5	98.9 ± 32.3	0.009
Insulin (U/L)	9.1 ± 5.9	5.9 ± 3.4	6.9 ± 4.5	<0.001
HOMA-IR	2.5 ± 2.2	1.4 ± 0.9	1.8 ± 1.6	<0.001
Total cholesterol (mg/dL)	182.9 ± 37.1	186.6 ± 34.9	192.3 ± 40.1	0.1
HDL cholesterol (mg/dL)	49.9 ± 12.1	54.3 ± 14.4	55.5 ± 12.2	<0.001
LDL cholesterol (mg/dL)	112.4 ± 33.3	113.7 ± 29.9	118.4 ± 35.9	0.29
Triglycerides (mg/dL)	104.4 ± 55.9	98.9 ± 51.5	92.1 ± 36.3	0.13
Haemoglobin	13.9 ± 1.4	14.1 ± 1.4	13.9 ± 1.3	0.61
Creatinine	0.8 ± 0.3	0.8 ± 0.2	0.8 ± 0.1	0.11
AST	23.6 ± 12.6	26.9 ± 22.4	24.9 ± 10.5	0.14
ALT	21.1 ± 12.4	22.5 ± 22.3	19.2 ± 10.7	0.81
GGT	18.5 ± 11.7	20.1 ± 14.2	29.9 ± 24.9	0.15

CI: confidence interval. BMI: body mass index; SMI: skeletal muscle index; FTSS: five-times-sit-to-stand; SPPB: Short Physical Performance Battery; HOMA-IR: homeostatic model assessment-insulin resistance; HDL: high-density lipoprotein; LDL: low-density lipoprotein; AST: aspartate amino transferase; ALT: alanine amino transferase; GGT: gamma-glutamyl transferase. All values: mean ± standard deviation (mean ± SD) for continuous variables. * *p*-value for coefficient trend.

**Table 4 ijerph-18-12979-t004:** Sociodemographic, lifestyle-related, anthropometric, sarcopenia-related, and laboratory biomarker characteristics according to the different sarcopenia-related categories, European Working Group on Sarcopenia in Older People 2019 (EWGSOP2) using skeletal muscle index (SMI) to define muscle mass. The *Salus in Apulia* Study (*n* = 740).

EWGSOP2	No Sarcopenia	Probable Sarcopenia	Sarcopenia + Severe Sarcopenia	*p* *
**Frequency**	*n* = 539 73% (95% CI: 65–76%)	*n* = 172 23% (95% CI: 20–27%)	*n* = 29 4% (95% CI: 3–5%)	
**Sociodemographic and lifestyle-related variables**				
Age (years)	75.5 ± 5.9	75.3 ±5.7	76.4 ± 6.4	0.77
Females	281 (52%)	98 (57%)	19 (66%)	0.50
Education (years)	6.8 ± 4.2	7.1 ± 4.3	7.0 ± 5.5	0.43
Smoking (yes, %)	23 (4%)	4 (2%)	2 (7%)	0.73
Physical activity	2 ± 1	1 ± 1	2 ± 1	<0.001
**Anthropometric variables**				
Height (cm)	157.4 ± 8.4	156.7 ± 9.1	158.4 ± 9.4	0.84
Weight (kg)	72.84 ± 13.80	75 ± 14	62.5 ± 10.3	0.14
BMI (kg/m^2^)	29.4 ± 5	30.3 ± 4.8	24.9 ± 3.2	0.12
**Sarcopenia assessment**				
SMI (kg/m^2^)	8.7 ± 1.6	8.9 ± 1.6	6.9 ± 1.2	0.01
FTSS Test	4 ± 0	2 ± 1	2 ± 1	<0.001
SPPB	11 ± 2	8 ± 3	9 ± 2	<0.001
**Laboratory biomarkers**				
Glucose (mg/dL)	103.2 ± 24.5	105.8 ± 29.1	101.2 ± 35.9	0.59
Insulin (U/L)	8.6 ± 5.6	9.3 ± 6.3	7.4 ± 4.8	0.74
HOMA-IR	2.3 ± 2.1	2.5 ± 1.9	1.9 ± 1.7	0.83
Total cholesterol (mg/dL)	181.9 ± 36.9	186.8 ± 35.8	198.4 ± 42.4	0.01
HDL cholesterol (mg/dL)	50.9 ± 12.9	48.9 ± 10.7	56.3 ± 12.5	0.98
LDL cholesterol (mg/dL)	111.7 ± 33.2	114.6 ± 32.3	122.7 ± 38.4	0.07
Triglycerides (mg/dL)	100.6 ± 54.9	112.3 ± 55.5	98.2 ± 38.4	0.12
Haemoglobin	13.9 ± 1.4	14.1 ± 1.5	13.9 ± 1.4	0.21
Creatinine	0.8 ± 0.2	0.8 ± 0.3	0.8 ± 0.1	0.80
AST	23.9 ± 14.1	24.2 ± 13.3	25.5 ± 11.6	0.54
ALT	20.7 ± 11.6	22.7 ± 18.5	19.9 ± 11.9	0.34
GGT	18.4 ± 11.5	19.3 ± 13.1	22.1 ± 27.9	0.13

CI: confidence interval. BMI: body mass index; FTSS: five-times-sit-to-stand; SPPB: Short Physical Performance Battery; HOMA-IR: homeostatic model assessment-insulin resistance; HDL: high-density lipoprotein; LDL: low-density lipoprotein; AST: aspartate amino transferase; ALT: alanine amino transferase; GGT: gamma-glutamyl transferase. All values: mean ± standard deviation (mean ± SD) for continuous variables. * *p*-value for coefficient trend.

**Table 5 ijerph-18-12979-t005:** Sociodemographic, lifestyle-related, anthropometric, sarcopenia-related, and laboratory biomarker characteristics according to the different sarcopenia-related categories, European Working Group on Sarcopenia in Older People 2019 (EWGSOP2) using appendicular skeletal muscle index (ASMI) to define muscle mass. The *Salus in Apulia* Study (*n* = 740).

EWGSOP2	No Sarcopenia	Probable-Sarcopenia	Sarcopenia + Severe Sarcopenia	*p* *
**Frequency**	*n* = 539 73% (95% CI: 69–76%)	*n* = 175 24% (95% CI: 21–27%)	*n* = 26 4% (95% CI: 2–5%)	
**Sociodemographic and lifestyle-related variables**				
Age (years)	75.5 ± 5.9	75.5 ± 5.8	75.15 ± 5.54	0.91
Females	281 (52.1)	108 (61.71)	9 (34.62)	0.83
Education (years)	6.78 ± 4.18	6.90 ± 4.30	8.38 ± 5.37	0.17
Smoking (yes, %)	23 (4.3)	5 (2.86)	1 (3.85)	0.51
Physical activity	2.12 ± 0.71	1.49 ± 0.86	1.72 ± 0.68	<0.001
**Anthropometric variables**				
Height (cm)	157.4 ± 8.4	156.3 ± 9.2	161.2 ± 7.4	0.73
Weight (kg)	72.8 ± 13.8	74.7 ± 13.8	59.9 ± 8.8	0.09
BMI (kg/m^2^)	29.4 ± 4.9	30.5 ± 4.5	22.9 ± 2.1	0.04
**Sarcopenia assessment**				
ASMI (kg/m^2^)	8.7 ± 1.6	8.7 ± 1.7	7.9 ± 1.4	0.22
FTSS Test	4 ± 0	1.6 ± 1.2	1.8 ± 1.2	<0.001
SPPB	11.1 ± 1.7	7.9 ± 2.7	8.4 ± 2.9	<0.001
**Laboratory biomarkers**				
Glucose (mg/dL)	103.2 ± 24.5	105.5 ± 28.7	102.9 ± 38.9	0.49
Insulin (U/L)	8.6 ± 5.6	9.6 ± 6.3	5.1 ± 2.4	0.73
HOMA-IR	2.3 ± 2.1	2.6 ± 1.9	1.4 ± 1.5	0.83
Total cholesterol (mg/dL)	181.9 ± 36.9	188.8 ± 36.5	16.1 ± 40.1	0.06
HDL cholesterol (mg/dL)	50.9 ± 12.9	49.5 ± 10.9	52.9 ± 12.4	0.61
LDL cholesterol (mg/dL)	111.7 ± 33.2	115.9 ± 32.8	114.8 ± 36.6	0.18
Triglycerides (mg/dL)	100.6 ± 54.9	113.2 ± 55.8	90.8 ± 28.6	0.17
Haemoglobin	13.9 ± 1.4	14.1 ± 1.5	14.4 ± 1.2	0.08
Creatinine	0.8 ± 0.3	0.8 ± 0.3	0.8 ± 0.1	0.86
AST	23.8 ± 14.1	24.4 ± 13.4	24.6 ± 11.1	0.62
ALT	20.7 ± 11.6	22.6 ± 18.5	20.2 ± 11.7	0.31
GGT	18.4 ± 11.5	20.1 ± 16.7	17.4 ± 10.7	0.41

CI: confidence interval; BMI: body mass index; FTSS: five-times-sit-to-stand; SPPB: Short Physical Performance Battery; HOMA-IR: homeostatic model assessment-insulin resistance; HDL: high-density lipoprotein; LDL: low-density lipoprotein; AST: aspartate amino transferase; ALT: alanine amino transferase; GGT: gamma-glutamyl transferase. All values: mean ± standard deviation (mean ± SD) for continuous variables. * *p*-value for coefficient trend.

## Data Availability

The datasets used and/or analyzed during the current study are available from the corresponding author on reasonable request.

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
