# Peer review of "Impact of Different Operational Definitions of Sarcopenia on Prevalence in a Population-Based Sample: The Salus in Apulia Study"

_ijerph, 2021, doi:10.3390/ijerph182412979_

Round 1

Reviewer 1 Report

I was pleased to read the article "Impact of Different Operational Definitions of Sarcopenia on Prevalence in a Population-based Sample: The Salus in Apulia Study ". Sarcopenia is defined as a progressive and systemic skeletal muscle disease that is associated with increased likelihood of activity problems and adverse consequences (including mortality). 

Differences in the definition of sarcopenia and the threshold values of sarcopenia parameters have limited the benefits obtained from the studies. To reach a global consensus for its standard definition and diagnosis, most recently, a revised EWGSOP consensus (EWGSOP2) has been published.

Although the aim of the study is worthy of investigation there are major flaws that should be revised substantially.  Please find my comments below."

  • In the EWGSOP2 definition, it was stated that the primary parameter of sarcopenia is low muscle strength. EWGSOP2 has also proposed standardized thresholds for sarcopenia parameters. It has been suggested that using normative reference values of the specific population, if any, can be used. In studies conducted in the Turkish population, specific cut-off values have been reported. (Bahat G, Aydin CO, Tufan A, Karan MA, Cruz-Jentoft AJ. Muscle strength cutoff values calculated from the young reference population to evaluate sarcopenia in Turkish population. Aging Clin Exp Res. 2021 Oct;33(10):2879-2882.)

 (Bahat G, Tufan A, Tufan F, Kilic C, Akpinar TS, Kose M, Erten N, Karan MA, Cruz-Jentoft AJ. Cut-off points to identify sarcopenia according to European Working Group on Sarcopenia in Older People (EWGSOP) definition. Clin Nutr. 2016 Dec;35(6):1557-1563.)

  • EWGSOP2 also stated that total skeletal muscle mass or appendicular skeletal muscle mass can be adjusted for body weight or body mass index, not just height squared. SMMI adjusted for height2 reflects lower muscle mass in people with a lower weight. (Yilmaz O, Bahat G. Muscle mass adjustment method affects the association of sarcopenia and sarcopenic obesity with metabolic syndrome. Geriatr Gerontol Int. 2019 Mar;19(3):272.)

  • SMMI adjusted for BMI is recommended by FNIH and also other studies (Bahat G, Tufan A, Kilic C, Aydın T, Akpinar TS, Kose M, Erten N, Karan MA, Cruz-Jentoft AJ. Cut-off points for height, weight and body mass index adjusted bioimpedance analysis measurements of muscle mass with use of different threshold definitions. Aging Male. 2020 Dec;23(5):382-387.)  (Bahat G, Tufan A, Kilic C, Öztürk S, Akpinar TS, Kose M, Erten N, Karan MA, Cruz-Jentoft AJ. Cut-off points for weight and body mass index adjusted bioimpedance analysis measurements of muscle mass. Aging Clin Exp Res. 2019 Jul;31(7):935-942.)

  • BMI-adjusted SMMI is correlated with functionality better. (Bahat G, Kilic C, Ilhan B, Karan MA, Cruz-Jentoft A. Association of different bioimpedanciometry estimations of muscle mass with functional measures. Geriatr Gerontol Int. 2019 Jul;19(7):593-597.)

  • It assigns the discriminating role to muscle function, when distinguishing non-sarcopenic, probably sarcopenic and sarcopenic subjects, instead of muscle mass.

  • Not only muscle function, but muscle function and/or muscle strength would be a more accurate description.

  • The category of probable sarcopenia was defined as having low muscle strength, and the category of sarcopenia was defined by the presence of low muscle mass.

  • If there is low muscle mass and/or low muscle function together with low muscle strength, it was termed as confirmed sarcopenia.

  • Table1- It may be more appropriate to specify SMI and ASMI values separately for men and women.

  • The change from muscle mass to muscle strength in the EWGSOP2 algorithm to diagnose sarcopenia was mostly motivated by feasibility rather than by strictly scientific arguments.

  • Here it is said that muscle strength is recommended for ease of use rather than scientific arguments. However, if sarcopenia is expressed as functional muscle failure, muscle strength is the most reliable measure of muscle function rather than mass. 

Author Response

Firstly, we would like to thank the reviewer for their suggestions and the precious time spent reviewing our manuscript.

In the EWGSOP2 definition, it was stated that the primary parameter of sarcopenia is low muscle strength. EWGSOP2 has also proposed standardized thresholds for sarcopenia parameters. It has been suggested that using normative reference values of the specific population, if any, can be used. In studies conducted in the Turkish population, specific cut-off values have been reported. (Bahat G, Aydin CO, Tufan A, Karan MA, Cruz-Jentoft AJ. Muscle strength cutoff values calculated from the young reference population to evaluate sarcopenia in Turkish population. Aging Clin Exp Res. 2021 Oct;33(10):2879-2882.)

 (Bahat G, Tufan A, Tufan F, Kilic C, Akpinar TS, Kose M, Erten N, Karan MA, Cruz-Jentoft AJ. Cut-off points to identify sarcopenia according to European Working Group on Sarcopenia in Older People (EWGSOP) definition. Clin Nutr. 2016 Dec;35(6):1557-1563.)

We agree with the referee, and we thank them for raising this issue. To the best of our knowledge, there are no specific muscle strength cut-offs for the Italian older population. We have clarified this point in the limitation of our study (page 13 “Moreover, we did not use specific cut-off for the Italian older population because, to the best of our knowledge, there are no studies on this subject, unlike other populations[47].”)

EWGSOP2 also stated that total skeletal muscle mass or appendicular skeletal muscle mass can be adjusted for body weight or body mass index, not just height squared. SMMI adjusted for height2 reflects lower muscle mass in people with a lower weight. (Yilmaz O, Bahat G. Muscle mass adjustment method affects the association of sarcopenia and sarcopenic obesity with metabolic syndrome. Geriatr Gerontol Int. 2019 Mar;19(3):272.)

SMMI adjusted for BMI is recommended by FNIH and also other studies (Bahat G, Tufan A, Kilic C, Aydın T, Akpinar TS, Kose M, Erten N, Karan MA, Cruz-Jentoft AJ. Cut-off points for height, weight and body mass index adjusted bioimpedance analysis measurements of muscle mass with use of different threshold definitions. Aging Male. 2020 Dec;23(5):382-387.)  (Bahat G, Tufan A, Kilic C, Öztürk S, Akpinar TS, Kose M, Erten N, Karan MA, Cruz-Jentoft AJ. Cut-off points for weight and body mass index adjusted bioimpedance analysis measurements of muscle mass. Aging Clin Exp Res. 2019 Jul;31(7):935-942.)

BMI-adjusted SMMI is correlated with functionality better. (Bahat G, Kilic C, Ilhan B, Karan MA, Cruz-Jentoft A. Association of different bioimpedanciometry estimations of muscle mass with functional measures. Geriatr Gerontol Int. 2019 Jul;19(7):593-597.)

We agree with the referee, but we established height2 approach due to the national data available and since it is covered in both consensus. However, we felt it was important to enrich our discussion with this interesting point thanks to the literature references suggested by the reviewer (page 11).

It assigns the discriminating role to muscle function, when distinguishing non-sarcopenic, probably sarcopenic and sarcopenic subjects, instead of muscle mass.

Not only muscle function, but muscle function and/or muscle strength would be a more accurate description.

As suggested, we have replaced the wording throughout the text.

The category of probable sarcopenia was defined as having low muscle strength, and the category of sarcopenia was defined by the presence of low muscle mass.

If there is low muscle mass and/or low muscle function together with low muscle strength, it was termed as confirmed sarcopenia.

On page 4, we modified the phrase as follow: “The category of probable sarcopenia was defined as having low muscle strength, and the category of sarcopenia defined by the presence of both low muscle mass and low muscle strength”

Table1- It may be more appropriate to specify SMI and ASMI values separately for men and women.

 We implemented Table1 with the information requested.

 The change from muscle mass to muscle strength in the EWGSOP2 algorithm to diagnose sarcopenia was mostly motivated by feasibility rather than by strictly scientific arguments.

 Here it is said that muscle strength is recommended for ease of use rather than scientific arguments. However, if sarcopenia is expressed as functional muscle failure, muscle strength is the most reliable measure of muscle function rather than mass. 

The referee is right. We added this sentence: “Nevertheless, since sarcopenia is expressed as functional muscle failure [1], it has also been stated that muscle strength is the most reliable measure of muscle function rather than muscle mass [24]” (Please see page 11, discussion section)

Reviewer 2 Report

The aim of the study was to investigate the impact of these different operational definitions on sarcopenia prevalence in a representative population-based sample. Although, this is a very great study, I have several major revisions. 

-Introduction: what is the hypothesis of study? Please, to clarify it.

-Methods: the authors could add the concordance/agreement between the EGWSOP1 vs EGWSOP2.

-Results: Please, to reduce the informations that has already been added in the tables.

-Discussion: ok

-Conclusion: please to remove "

Since sarcopenia is considered as one of the consequences of the aging progress, an early identification of subjects at risk can be an effective prevention strategy, that is a scientific and clinical priority for both researchers and clinicians. However, solid prospec-tive data on the potential health consequences of such conditions are needed, and a re-finement of suitable tools to identify conditions posing increased health risks."

Author Response

Firstly, we would like to thank the reviewer for having accurately reviewed our paper and for the helpful advice given.

-Introduction: what is the hypothesis of study? Please, to clarify it.

The reviewer is right. We added this period in the introduction “In the present study, data from the general population recruited in Southern Italy for the ongoing long-term Salus in Apulia study were used were used to estimate the prevalence of sarcopenia, assuming a substantial difference in percentage terms according to the two consensuses.” (page 2)

Moreover, we added this sentence to the conclusion: “In addition, use of the two different algorithms did not result in a substantially different percentage of subjects with sarcopenia in our study population.” (page 13)

-Methods: the authors could add the concordance/agreement between the EGWSOP1 vs EGWSOP2.

We thank the reviewer for raising this issue. On page 4, paragraph “Algorithms for the operational definitions of sarcopenia” we stated that the two consensuses share the definition of severe sarcopenia (“As with the EWGSOP1 definition, the category of severe sarcopenia was assigned when low muscle strength, low muscle mass, and low physical performance was present”). Previously, we also stated that the SPPB cut-off is the same in both consensuses (“A cutoff value of 8 in the SPBB score was considered to indicate low physical performance, in accordance with the both the EWGSOP panels”).

-Results: Please, to reduce the informations that has already been added in the tables.

We have reduced the results section as recommended by the referee (please check pages 4-11)

-Conclusion: please to remove "Since sarcopenia is considered as one of the consequences of the aging progress, an early identification of subjects at risk can be an effective prevention strategy, that is a scientific and clinical priority for both researchers and clinicians. However, solid prospective data on the potential health consequences of such conditions are needed, and a refinement of suitable tools to identify conditions posing increased health risks."

As suggested, we removed this part from the manuscript.

Round 2

Reviewer 1 Report

The authors have addressed all concerns raised by the reviewer. I would recommend the publication of this work in the journal 'International Journal of Environmental Research and Public Health.'

Reviewer 2 Report

In the Results section: the authors could add the concordance/agreement between the EGWSOP1 vs EGWSOP2. The authors explain it in Methods, but no data were provided in the results.